# Finding Biological Plausibility for Adversarially Robust Features via Metameric Tasks

## Abstract

Recent work suggests that feature constraints in the training datasets of deep neural networks (DNNs) drive robustness to adversarial noise (Ilyas et al., 2019). The representations learned by such adversarially robust networks have also been shown to be more human perceptually-aligned than non-robust networks via image manipulations (Santurkar et al., 2019; Engstrom et al., 2019). Despite appearing closer to human visual perception, it is unclear if the constraints in robust DNN representations match biological constraints found in human vision. Human vision seems to rely on texture-based/summary statistic representations in the periphery, which have been shown to explain phenomena such as crowding (Balas et al., 2009) and performance on visual search tasks (Rosenholtz et al., 2012). To understand how adversarially robust optimizations/representations compare to human vision, we performed a psychophysics experiment using a metamer task similar to Freeman & Simoncelli (2011); Wallis et al. (2019); Deza et al. (2017) where we evaluated how well human observers could distinguish between images synthesized to match adversarially robust representations compared to non-robust representations and a texture synthesis model of peripheral vision (Texforms (Long et al., 2018)). We found that the discriminability of robust representation and texture model images decreased to near chance performance as stimuli were presented farther in the periphery. Moreover, performance on robust and texture-model images showed similar trends within participants, while performance on non-robust representations changed minimally across the visual field. These results together suggest that (1) adversarially robust representations capture peripheral computation better than non-robust representations and (2) robust representations capture peripheral computation similar to current state-of-the-art texture peripheral vision models. More broadly, our findings support the idea that localized texture summary statistic representations may drive human invariance to adversarial perturbations and that the incorporation of such representations in DNNs could give rise to useful properties like adversarial robustness.

## 1  Introduction

Texture-based summary statistic models of the human periphery have been shown to explain key phenomena such as crowding (Balas et al., 2009; Freeman & Simoncelli, 2011) and performance on visual search tasks (Rosenholtz et al., 2012) when used to synthesize feature-matching images. These analysis-by-synthesis models have also been used to explain mid-level visual computation (*e.g.* V2) via perceptual discrimination tasks on images for humans and primates (Freeman & Simoncelli, 2011; Ziemba et al., 2016; Long et al., 2018).

Submitted to 3rd Workshop on Shared Visual Representations in Human and Machine Intelligence (SVRHM 2021) of the Neural Information Processing Systems (NeurIPS) conference.

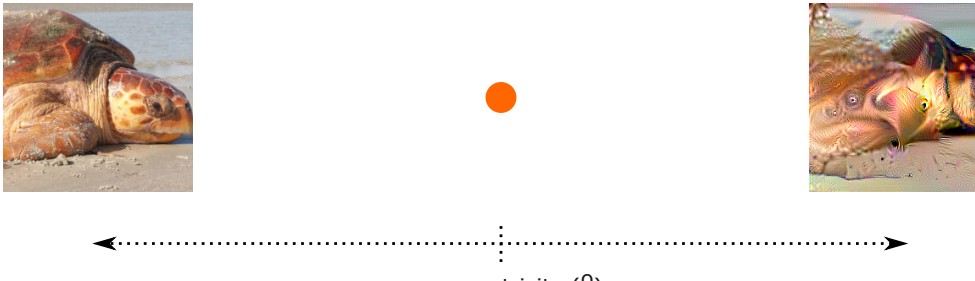

eccentricity ($^o$)

Figure 1: A sample un-perturbed (left) and synthesized adversarially robust (right) image are shown peripherally. When a human observer fixates at the orange dot (center), both images – now placed away from the fovea – are perceptually indistinguishable to each other (i.e. *metameric*). In this paper we investigate if there is a relationship between peripheral representations in humans and learned representations of adversarially trained networks in machines in an analysis-by-synthesis approach. We psychophysically test this phenomena over a variety of images synthesized from an adversarially trained network, a non-adversarially trained network, and a model of peripheral computation as we manipulate retinal eccentricity over 12 humans subjects.

While summary statistic models can succeed at explaining peripheral computation in humans, they fail to explain foveal computation and core object recognition that involve other representational strategies (Logothetis et al., 1995; Riesenhuber & Poggio, 1999; DiCarlo & Cox, 2007; Hinton, 2021). Modelling foveal vision with deep learning indeed has been the focus of nearly all object recognition systems in computer vision (as machines do not have a periphery) (LeCun et al., 2015; Schmidhuber, 2015). Despite their success, however, they are vulnerable to adversarial perturbations. This phenomena indicates: 1) a critical failure of current artificial systems (Goodfellow et al., 2014; Szegedy et al., 2013); and 2) a perceptual mis-alignment of such systems with humans (Golan et al., 2019; Feather et al., 2019; Firestone, 2020; Geirhos et al., 2021; Funke et al., 2021) – with some exceptions (Elsayed et al., 2018). Indeed, there are many strategies to alleviate these sensitivities to perturbations, such as data-augmentation (Rebuffi et al., 2021), biologically-plausible inductive biases (Dapello et al., 2020; Reddy et al., 2020; Jonnalagadda et al., 2021), and adversarial training (Tsipras et al., 2018; Madry et al., 2017). This last strategy in particular (adversarial training) is popular, but has been criticized as being non-biologically plausible – despite yielding some perceptually aligned images when inverting their representations (Engstrom et al., 2019; Santurkar et al., 2019).

We know machines do not have peripheral computation, yet are susceptible to a type of adversarial attacks that humans are not. We hypothesize that object representation arising in human peripheral computation holds a critical role for high level robust vision in perceptual systems, but testing this has not been done. Inspired by recent works that test have tested summary statistic models via metameric discrimination tasks (Deza et al., 2017; Wallis et al., 2016, 2017, 2019), we can evaluate how well the adversarially robust CNN model approximates the types of computations present in human peripheral vision with a set of rigorous psychophysical experiments with respect to synthesized stimuli (Figure 1). We evaluated the rates of human perceptual discriminability as a function of retinal eccentricity across the synthesized stimuli from an adversarially trained network vs synthesized stimuli from models of mid-level/peripheral computation. If the decay rates of perceptual discriminability are similar across stimuli, then it suggests that the transformations learned in an adversarially trained network are isomorphic to the transformations done by models of peripheral computation – and thus, to the human visual system.

## 2 Human Psychophysics: Discriminating between stimuli as a function of retinal eccentricity

We designed two human psychophysical experiments: the first was a an oddity task similar to Wallis et al. (2016), and the second was a matching, two-alternative forced choice task (2AFC). Two different tasks were used to evaluate how subjects viewed synthesized images both only in the periphery (oddity) and those they saw in the fovea (matching 2AFC). The oddity task consisted of finding the oddball stimuli out of a series of 3 stimuli shown peripherally one after the other (100ms)

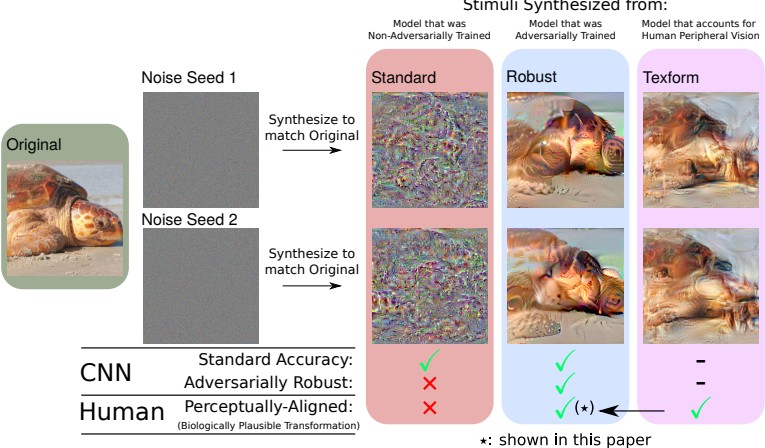

Figure 2: A sub-collection of synthesized stimuli used in our experiments that show differences across (columns) and within (rows) perceptual models. The original stimuli is shown on the left, with two parallel Noise Seeds that give rise to synthesized samples for the Standard, Robust and Texform stimuli. Critically, an adversarially trained network – which was used to synthesize the Robust stimuli (Engstrom et al., 2019) – has implicitly learned to encode a structural prior with localized texture-like distortions similar to the physiologically motivated Texforms that account for several phenomena of *human peripheral computation* (Freeman & Simoncelli, 2011; Rosenholtz et al., 2012; Long et al., 2018). However, Standard stimuli, which are images synthesized from a network with Regular (Non-Adversarial) training have no resemblance to the original sample.

masked by empty intervals (500ms) while holding center fixation. Chance for the oddity task was 1 out of 3 (33.3%). The matching 2AFC task consisted of viewing a stimulus in the fovea (100ms) and then matching it to two candidate templates in the visual periphery (100 ms) while holding fixation. A 1000 ms mask was used in this experiment and chance was 50%.

We used 3 types of stimuli in our experiments: Standard stimuli which were synthesize by a non-adversarially (standard) trained networks, Robust stimuli which are synthesized by an adversarially trained stimuli, and Texform stimuli which are synthesized by a model of peripheral and mid-ventral computation (Figure 2). More information on stimuli synthesis can be seen in Appendix A.

For both experiments, we also had interleaved trials where observers had to engage in a Original stimuli vs Synthesized stimuli task, or a Synthesized stimuli vs Synthesized stimuli discrimination task (two stimulus pairs synthesized from *different* noise seeds to match model representations). The goal of these experimental variations (called *'stimulus roving'*) was two-fold: 1) to add difficulty to the tasks thus reducing the likelihood of ceiling effects; 2) to gather two psychometric functions per family of stimuli, which portrays a better description of each stimuli's perceptual signatures.

We had 12 participants complete both the oddity and matching 2AFC experiments as shown in Figure 3. The oddity task was always performed first so that subjects would never have foveated on the images before seeing them in the periphery. We had two stimulus conditions (1) robust & standard model images and (2) texforms. Condition 1 consisted of the inverted representations of the adversarially robust and standard-trained models. The two model representations were randomly interleaved since they were synthesized with the same procedure. Condition 2 consisted of texforms synthesized with a fixed and perceptually optimized fixation and scaling factor which yielded images closest in structure to the robust representations at foveal viewing (robust features have no known fixation and scaling – which is why we evaluate multiple points in the periphery. Recall Figure 5). We randomly assigned the order in which participants saw the different stimuli.

The main results of our 2 experiments can be found in Figure 4, where we show how well Humans can discriminate per type of stimuli class and task. Mainly Human observers achieve near perfect discrimination rates for the Standard stimuli wrt to their original references, but near chance levels when discriminating to another synthesized sample. This occurs for both experimental paradigms (Oddity + 2AFC), suggesting that the network responsible for encoding standard stimuli is a poor model of human peripheral vision given no interaction with retinal eccentricity.

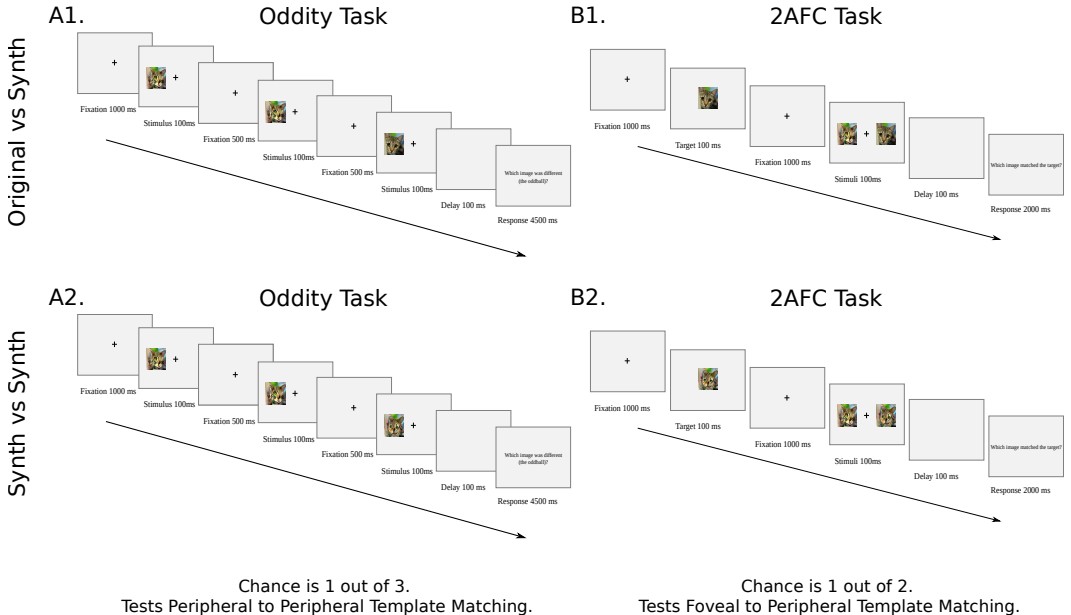

Figure 3: A schematic of the two human psychophysics experiments conducted in our paper. The first (A1.,A2.) illustrates an Oddity task where observers must determine the 'oddball' stimuli without moving their eyes for brief presentation times (100 ms) which do not allow for eye-movements or feedback processing. The second experiment (B1.,B2.) shows the 2 Alternative Forced Choice (2AFC) Matching Tasks where observers must match the foveal template to 2 potential candidates on the left or right of the image. All trials are done while observers are instructed to remain fixating at the center of the image. Differences across rows indicate the type of interleaved trials shown to the observers: (1) Original vs Synthesized, and (2) Synthesized vs Synthesized.

However, we observe that Humans show similar perceptual discriminability rates for Robust and Texform stimuli – and that these vary in a similar way as a function of retinal eccentricity. Indeed, for both of these stimuli their perceptual discrimination rates appear to follow a sigmoidal decay-like curve when comparing the stimuli to the original, and also between synthesized samples. The similarity between the blue and magenta curves from Figure 4 suggests that if the texform stimuli do capture some aspect of peripheral computation, then – by transitivity – so do the adversarial stimuli which were rendered from an adversarially trained network. These results empirically verify our initial hypothesis that adversarially trained networks encode a similar set of transformations as the human visual periphery. A superposition of these results in reference to the Robust stimuli for a better interpretation can also be seen in Figure 4 (B.).

## 3   Discussion

We found that stimuli synthesized from an adversarially trained (and thus robust) network are metameric to the original stimuli in the further periphery (slightly above $30 \deg$) for both Oddity and 2AFC Matching tasks. However, more important than deriving a critical eccentricity for metameric guarantees across stimuli in Humans – we found a surprisingly similar pattern of results in terms of how perceptual discrimination interacts with retinal eccentricity when comparing the adversarially trained network's robust stimuli with classical models of peripheral computation and V2 encoding (mid-level vision) that were used to render the texform stimuli (Freeman & Simoncelli, 2011; Long et al., 2018; Ziemba et al., 2016; Ziemba & Simoncelli, 2021). Further, this type of eccentricity-driven interaction does not occur for stimuli derived from non-adversarially trained (standard) networks.

More generally, now that we found that adversarially trained networks encode a similar class of transformations that occur in the visual periphery – how do we reconcile the fact that adversarial training is biologically *implausible* in humans? Recall from the work of Ilyas et al. (2019) that per-forming *standard training* on robust images yielded similar generalization and adversarial robustness

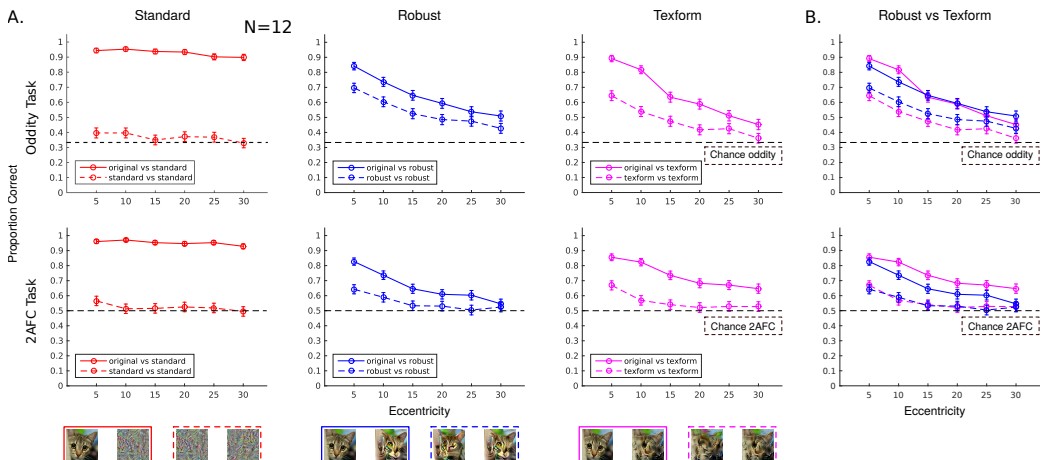

Figure 4: Pooled observer results of both psychophysical experiments are shown (top and bottom row). (A.) Left: we see that observers almost perfectly discriminate the original image from the standard stimuli, in addition to chance performance when comparing against synthesized stimuli. Critically there is no interaction of the standard stimuli with retinal eccentricity which suggests that the model used to synthesize such stimuli is a poor model of peripheral computation. Middle: Human observers do worse at discriminating the robust stimuli from the original as a function of eccentricity and also between synthesized robust samples. Given this decay in perceptual discriminability, it suggests that the adversarially trained model used to synthesize robust stimuli does capture aspects of peripheral computation. This effect is also seen with the texforms (Right) – which have been extensively used as stimuli from derived models that capture peripheral and V2-like computation. (B.) Superimposed human performance for Robust and Texform stimuli. Errorbars are computed via bootstrapping and represent the 95% confidence interval.

as performing adversarial training on standard images; how does this connect then to human learning if we assume a uniform learning rule in the fovea and the periphery?

We think the answer lies in the fact that as humans learn to perform object recognition, they not only fixate at the target image, but they also look around, and can eventually learn where to make a saccade given candidate object peripheral template – thus learning certain invariances when the object is placed both in the fovea and the periphery (Cox et al., 2005; Williams et al., 2008; Poggio et al., 2014; Han et al., 2020). This is an idea that dates back to Von Helmholtz (1867) as highlighted in Stewart et al. (2020) on the interacting mechanisms of foveal and peripheral vision in humans.

Altogether, this could suggest that spatially-uniform high-resolution processing is redundant and sub-optimal in the *o.o.d.* regime – as translation invariant adversarially-vulnerable CNNs have no foveated/spatially-adaptive computation. Counter-intuitively, the fact that our visual system *is* spatially-adaptive could give rise to a more robust encoding mechanism of the visual stimulus as observers can encode a distribution rather than a point as they move their center of gaze. Naturally, from all the possible types of transformations, the ones that are similar to those shown in this paper – which loosely resemble localized texture-computation – are the ones that potentially lead to a robust hyper-plane during learning for the observer (See Fig. 9; Appendix).

Future work is looking into reproducing the experiments carried out in this paper with a physiological component to explore temporal dynamics (MEG) and localization (fMRI) evoked from the stimuli. While it is not obvious if we will find a perceptual signature of the adversarial robust stimuli in humans, we think this novel stimuli and experimental paradigm presents a first step towards the road of linking what is known (and unknown) across texture representation, peripheral computation, and adversarial robustness in humans and machines.

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

# A    Synthesizing Stimuli as a window to Model Representation

Suppose we have the functions $g_{Adv}(\circ)$ and $g_{Standard}(\circ)$ that represent the adversarially trained and standard (non-adversarially) trained neural networks; how can we compare them to human peripheral computation if the function $g_{Human}(\circ)$ is computationally intractable?

One solution is to take an analysis-by-synthesis approach and to synthesize a collection of stimuli that match the feature response of the model we'd like to analyze – this is also known as feature inversion (Mahendran & Vedaldi, 2015; Feather et al., 2019). If the inverted features (stimuli) of two models are perceptually similar, then it is likely that the learned representations are also aligned. For example, if we'd like to know what is the stimuli $x'$ that produces the same response to the stimuli $x$ for a network $g'(\circ)$, we can perform the following minimization:

$$x' =_{x_0} [||g'(x) - g'(x_0)||_2] \tag{1}$$

In doing so, we find $x'$ which should be different from $x$ for a non-trivial solution. This is known as a metameric constraint for the stimuli pair $\{x, x_0\}$ wrt to the model $g'(\circ) : g'(x) = g'(x')$ s.t. $x \neq x'$ for a starting pre-image $x_0$ that is usually white noise in the iterative minimization of Eq.1. Indeed, for the adversarially trained network of Ilyas et al. (2019); Engstrom et al. (2019); Santurkar et al. (2019), we can synthesize robust stimuli wrt to the original image $x$ via:

$$\tilde{x} =_{x_0} [||g_{Adv}(x) - g_{Adv}(x_0)||_2] \tag{2}$$

which implies – if the minimization goes to zero – that:

$$||g_{Adv}(x) - g_{Adv}(\tilde{x})||_2 = 0 \tag{3}$$

Recalling the goal of this paper, we'd like to investigate if the following statement is true: *"a transformation resembling peripheral computation in the human visual system can closely be approximated by an adversarially trained network"*, which is formally translated as: $g_{Adv} \sim g_{Human}^{r_*}$ for some retinal eccentricity $(r_*)$, then from Eq. 3 we can also derive:

$$||g_{Human}^{r_*}(x) - g_{Human}^{r_*}(\tilde{x})||_2 = 0 \tag{4}$$

However, $g_{Human}(\circ)$ is computationally intractable, so how can we compute Eq.4? A first step is to perform a psychophysical experiment such that we find a retinal eccentricity $r_*$ at which human observers can not distinguish between the original and synthesized stimuli – thus behaviourally proving that the condition above holds, without the need to directly compute $g_{Human}$.

More generally, we'd like to compare the *psychometric functions* between stimuli generated from a standard trained network (standard stimuli), an adversarially trained network (robust stimuli), and a model that captures peripheral and mid-level visual computation (texform stimuli (Freeman & Simoncelli, 2011; Long et al., 2018)). Then we will assess how the psychometric functions vary as a function of retinal eccentricity. If there is significant overlap between psychometric functions between one model wrt the model of peripheral computation; then this would suggest that the transformations developed by such model are similar to those of human peripheral computation. We predict that this will be the case for the adversarially trained network ($g_{Adv}(\circ)$). Formally, for any model $g$, and its synthesized stimuli $x_g$ – as shown in Figure 2, we will define the psychometric function $\delta_{Human}$, which depends on the eccentricity $r$ as:

$$\delta_{Human}(g; r) = ||g_{Human}^r(x) - g_{Human}^r(x_g)||_2 \tag{5}$$

where we hope to find:

$$\delta_{Human}(g_{Adv}; r) = \delta_{Human}(g_{Texform}; r); \forall r. \tag{6}$$

## A.1    Standard and Robust Model Stimuli

To evaluate robust vs non-robust feature representations, we used the ResNet-50 models of Santurkar et al. (2019); Ilyas et al. (2019); Engstrom et al. (2019). We used their models so that our results could be interpreted in the context of their findings that features may drive robustness. Both models were trained on a subset of ImageNet (Russakovsky et al., 2015), termed Restricted ImageNet (Table 1). The benefit of Restricted ImageNet, stated by Ilyas et al.; Engstrom et al., is models can achieve better standard accuracy than on all of ImageNet. One drawback is that it is imbalanced across classes. Although the class imbalance was not problematic for comparing the adversarially robust model to

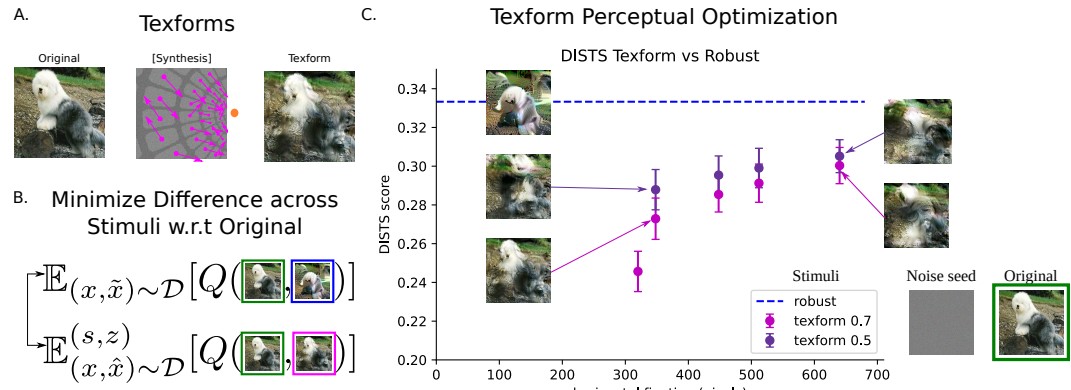

Figure 5: (A.) A cartoon depicting the texform generating process where log-polar receptive fields are used as areas over which localized texture synthesis is performed – imitating the type of texture-based computation found in the human periphery and area V2. (B.) The perceptual optimization framework where the goal is to find the set of texform parameters $(s_*, z_*)$ over which the loss is minimized to match the levels of distortions of the robust stimuli *before* performing human psychophysics. (C.) The texform perceptual optimization pipeline results show the DISTS scores (Ding et al., 2020) of texforms synthesized across different scaling factors and fixations points compared to adversarially robust stimuli synthesized from the same noise seed across 45 images (5 per RestrictedImageNet class selected randomly). Error bars indicate two standard errors from the mean.

317 standard-trained one, we did ensure that there was a nearly equal number of images per class when
318 selecting images for our stimulus set to avoid class effects in our experiment (i.e. people are better at
319 discriminating dog examples than fishes independent of the model training).

320 Using their readily available models, we synthesized robust and standard model stimuli using an
321 image inversion procedure (Mahendran & Vedaldi, 2015; Gatys et al., 2015; Santurkar et al., 2019;
322 Engstrom et al., 2019; Ilyas et al., 2019). We used gradient descent to minimize the difference
323 between the representation of the second-to-last network layer of a target image and an initial noise
324 seed as shown in Figure 6. Target images were randomly chosen from the test set of Restricted
325 ImageNet. We chose 100 target images for each of the 9 classes and synthesized a robust and standard
326 stimulus for 2 different noise seeds. 5 target images were later removed as they were gray-scale and
327 could not also be rendered as Texforms with the same procedure as the majority. All stimuli were
328 synthesized at a size of 256 by 256 pixels, this was equivalent to $7 \times 7$ degree of visual angle (d.v.a.)
329 when performing the psychophysical experiments.

## A.2 Texform Stimuli

331 Texforms (Long et al., 2018) are object-equivalent rendered stimuli from the Freeman & Simoncelli
332 (2011); Rosenholtz et al. (2012) models that break the metameric constraint to test for mid-level
333 visual representations in Humans. These stimuli – initially inspired by the experiments of Balas et al.
334 (2009) – preserve the coarse global structure of the image and its localized texture statistics (Portilla
335 & Simoncelli, 2000). Critically, we use the texform stimuli – *voiding the metameric constraint* – as a
336 perceptual control for the robust stimuli, as the texforms incarnate a sub-class of biologically-plausible
337 distortions that loosely resemble the mechanisms of human peripheral processing.

338 As the texform model has 2 main parameters which are the scaling factor $s$ and the simulated point
339 of fixation $z$, we must perform a perceptual optimization procedure to find the set of texforms $\hat{x}$
340 that match the robust stimuli $\tilde{x}$ as close as possible (w.r.t to the original image) *before* testing their
341 discriminability to human observers as a function of eccentricity. To do this, we used the accelerated
342 texform implementation of Deza et al. (2019) and generated 45 texforms with the *same* collection of
343 initial noise seeds as the robust stimuli to be used as perceptual controls. Similar to Deza & Konkle
344 (2020) we minimize the perceptual dissimilarity $\mathcal{Z}$ to find $(s_*, z_*)$ over this subset of images that we
345 will later use in the human psychophysics ($\sim 900$ texforms):

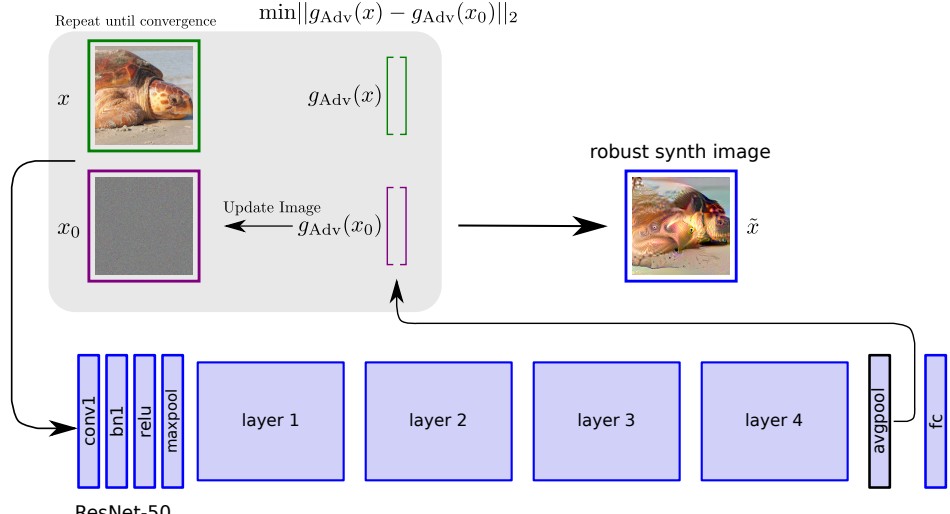

Figure 6: The Robust Image Synthesis pipeline: A noise image $x_0$ is passed through an adversarially trained ResNet-50 and the penultimate layer features $g_{Adv}(x_0)$ are matched wrt the original images' penultimate feature activation $g_{Adv}(x)$ via an L2 loss, and is repeated until convergence (Santurkar et al., 2019; Engstrom et al., 2019). Critically we use $g_{Adv}(\circ)$ as a summary statistic of peripheral processing in our experiments.

$$(s_*, z_*) =_{(s,z)} \mathcal{Z} = ||\mathbb{E}_{(x,\tilde{x})\sim\mathcal{D}}[Q(x,\tilde{x})] - \mathbb{E}^{(s,z)}_{(x,\hat{x})\sim\mathcal{D}}[Q(x,\hat{x})]||_2 \qquad (7)$$

for an image quality assessment (IQA) function $Q(\circ, \circ)$. We selected DISTS in our perceptual optimization setup given that it is the IQA metric that is most tolerant to texture-based transformations (Ding et al., 2020, 2021). A cartoon illustrating the texform rendering procedure, the perceptual optimization framework and the respective results can be seen in Figure 5. In our final experiments we used texforms rendered with a simulated scale of 0.5 and horizontal simulated point of fixation placed at 640 pixels. Critically, this value is *immutable* and texforms (like robust stimuli) will not vary as a function of eccentricity to provide a fair discriminability control in the human psychophysics. For a further discussion on texforms and their biological plausibility and/or synthesis procedure, please see Supplement B.2.

## B    Image Synthesis Details

| | | | | | **Classes** | | | | |
|---|---|---|---|---|---|---|---|---|---|
| **RIN** | Dog | Cat | Frog | Turtle | Bird | Primate | Fish | Crab | Insect |
| **IN** | 151-268 | 281-285 | 30-32 | 33-37 | 68-100 | 365-382 | 389-397 | 118-121 | 300-319 |

Table 1: Classes of RestrictedImageNet (**RIN**) and the corresponding ImageNet (**IN**) class ranges.

### B.1    Standard and Robust Stimuli

We used the publicly available code from Santurkar et al. (2019); Engstrom et al. (2019); Ilyas et al. (2019) found here to synthesize both standard and robust stimuli which where derived from a regularly and adversarially trained model respectively: `https://github.com/MadryLab/robust_representations`

A schematic that illustrates the robust stimuli rendering pipeline can be seen in Figure 6. Standard stimuli is generated with the same procedure, and number of iterations, but the network $g_{Adv}(\circ)$ is replaced with $g_{Standard}(\circ)$ instead.

## B.2 Texform Stimuli

Texform stimuli were synthesized using the publicly available code of Deza et al. (2019): `https://github.com/ArturoDeza/Fast-Texforms`

The following images (class:[image id's]) were removed as they did not converge:

- texform0: 0:[49],1:[9],2:[],3:[44],4:[],5:[],6:[10],7:[40],8:[].
- texform1: 0:[49],1:[9,44],2:[],3:[44],4:[],5:[],6:[10],7:[40],8:[]

In addition the following image id's were removed from our psychophysical analysis from the texform stimuli as they converged to the *exact* same image even when starting from different noise seeds. This was found while doing a post-hoc IQA analysis as the one shown in Figure **??**. These stimuli only occurred for classes 0 (dog) and 1 (cat):

- texform: 0:[22,25,26,27,29,93,94,95,96,97,98,99],1:[20,21,22,23,73,74]

We found that Standard and Robust stimuli did not have this identical convergence problem over the 900 rendered pairs (1800 stimuli in total for Standard and 1800 in total for Robust).

**Note 1a:** A common mis-conception is that Freeman & Simoncelli (2011)-derived stimuli (such as texforms) *do not* contain structural priors and only performs localized texture synthesis over smoothly overlapping log-polar receptive fields. This has been investigated with great detail in Wallis et al. (2016, 2017); Liu et al. (2016) that showed that without spectral constraints it is impossible to generate metameric images from non-stationary textures for the human observer when showing such stimuli in the visual periphery. For texforms the metameric constraint is purposely broken because we'd like to test how a specific biologically-plausible family of transformations (embodied through the synthesis procedure) interacts with eccentricity when the eccentricity-dependent and scaling factors texform parameters are fixed. See $(z_*, s_*)$ from Eq. 7.

**Note 1b:** The Freeman & Simoncelli (2011) synthesis model is not equivalent to the Portilla & Simoncelli (2000) synthesis model. The Freeman & Simoncelli (2011) is a super-ordinate synthesis model class that locally uses the Portilla & Simoncelli (2000) synthesis model over smoothly overlapping receptive fields in addition to adding a global structural prior. Texforms are rendered with the Freeman & Simoncelli (2011) model, by placing he simulated point of fixation *outside* the image (Long et al., 2018; Deza et al., 2019).

**Note 1c:** Usual texform rendering time is about 1 day per image, though the rendering procedure has been accelerated to the order of minutes as shown in Deza et al. (2019). We used their publicly available code in our experiments. Thus, it is worth noting that synthesizing texforms in the order of hundreds of thousands (or millions) for supervised learning experiments – has not been done before and is computationally expensive (may take months), which is why Figure 2 displays no information on texform-trained CNN's. This direction is current work.

**Note 2:** A first naive criticism to the selection of making texforms fixed and not varying as a function of eccentricity – given the model they were based on (Freeman & Simoncelli, 2011) – is that they will not create metameric stimuli. Our anticipated reply to this is three-fold, and partially aligned with the motivation of Long et al. (2018):

1. Our goal is *not* to make metameric stimuli out of texforms or robust stimuli, but to examine how perceptual discriminability rates of a *fixed stimuli* change as a function of retinal eccentricity. By checking if these perceptual decays are similar (which we show) we can connect both functions that give rise to these apparently un-related transformations (the stimuli). Recall Eq. 6.

2. Having a "metameric texform" that changes as a function of eccentricity would defeat the purpose of using it as a control in our experiments. Had this been the road taken, we would now have a control curve that will presumably be horizontal and at chance, providing no information about how the transformation that gives rise to the robust stimuli is linked to the texform transformation.

3. The goal of this paper is *not* to make a foveated metamer model that fools human observers similar to that of Freeman & Simoncelli (2011); Rosenholtz et al. (2012); Deza et al. (2017);

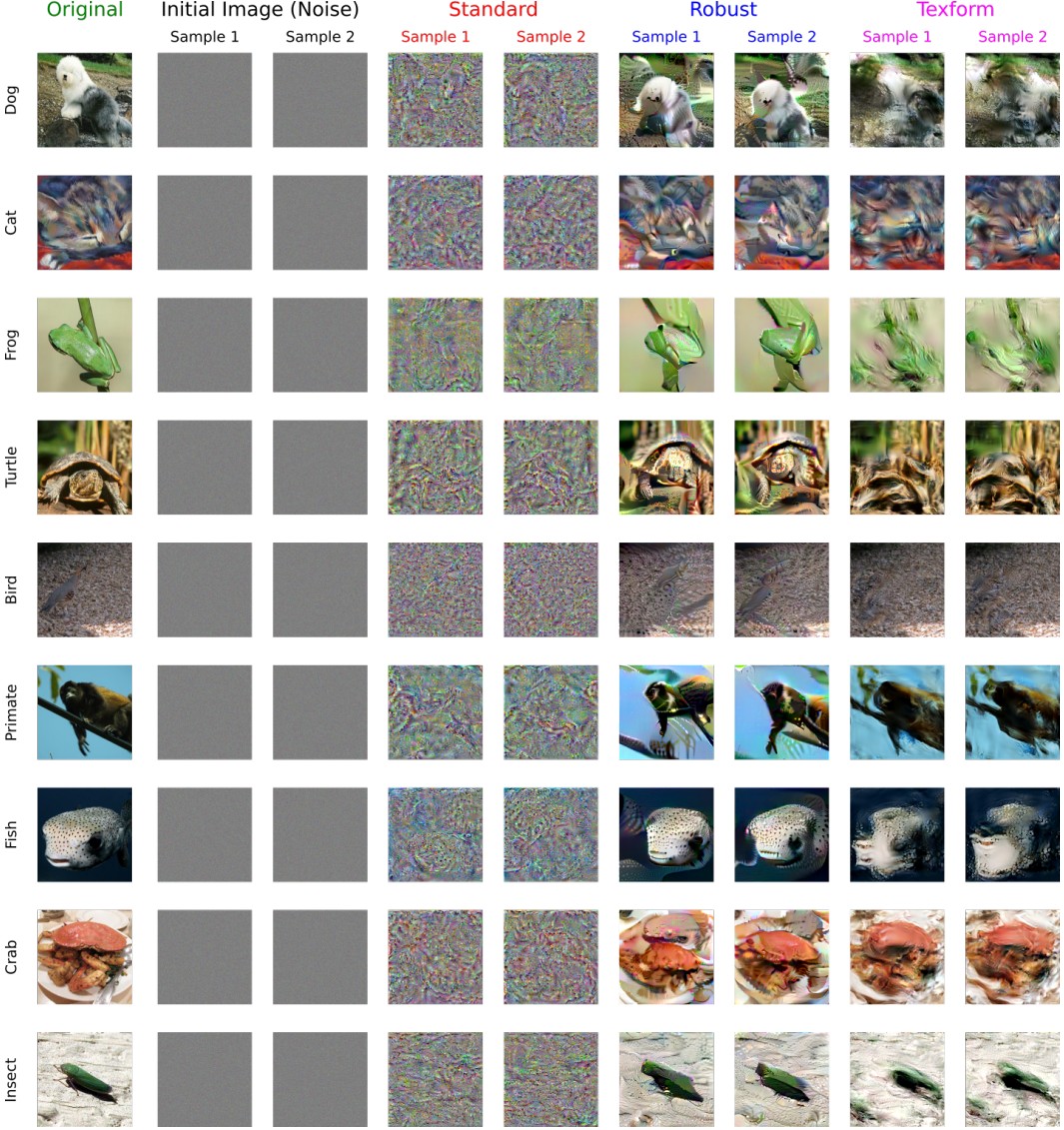

Figure 7: A collection of sample stimuli for each image class used in our experiments.

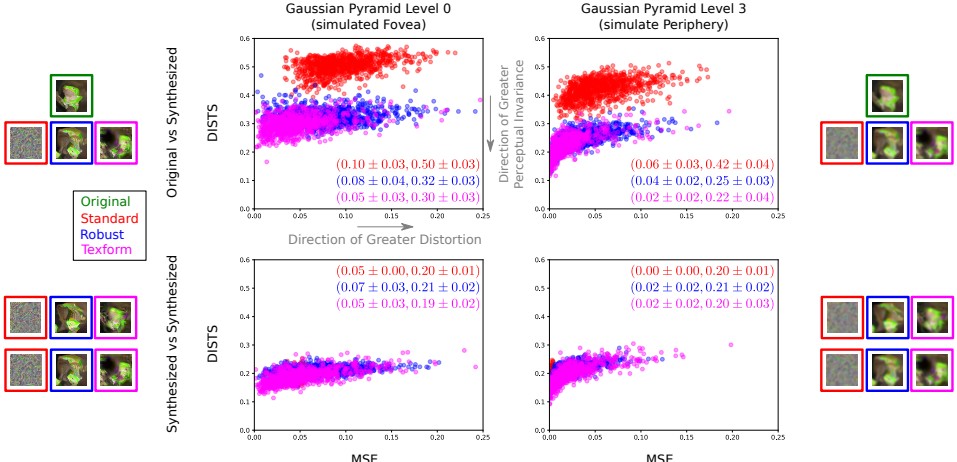

Figure 8: Here we evaluate how the different stimuli differ to each other wrt to the original (top row) or synthesized samples (bottom row) via two IQA metrics: DISTS and MSE. This characterization allows us to compare which model discards more information (MSE) while yielding a greater degree of model based perceptual invariance. We find that Texform and Robust stimuli are similar terms of both IQA scores, suggesting their models compute the same transformations. This is observed at the 0th level (simulated fovea) and 3rd level (simulated periphery) of the Gaussian Pyramid.

## C Simulated Fovea/Periphery Image Quality Assessment (IQA) across stimuli

Some distortions are more perceptually noticeable than others for human observers and deep neural networks (Berardino et al., 2017) – so how do we assess which model better accounts for peripheral computation, if there are many distortions (derived from the synthesized model stimuli) that can potentially yield the same perceptual sensitivity in a discrimination task?

Our approach consists of computing two IQA metrics (DISTS & MSE) over the entire psychophysical testing set over 2 opposite levels of a Gaussian Pyramid decomposition (Burt & Adelson, 1987). This procedure checks which stimuli presents the greatest distortion (MSE), and yet yields greater perceptual invariance (DISTS). A Gaussian Pyramid decomposition was selected as it stimulates the frequencies preserved given changes in human contrast sensitivity and cortical magnification factor from fovea to periphery (Anstis, 1974; Geisler & Perry, 1998). These two metrics were one that is texture-tolerant and perceptually aligned (DISTS), and another that is a non-perceptually aligned metric: Mean Square Error (MSE). Both IQA metrics were computed in pixel space for both the Original vs Synthesized and Synthesized vs Synthesized conditions.

Results are explained in Figure 8, where Standard Stimuli yields low perceptual invariance to the original image at both levels of the Gaussian Pyramid, but robust and texform stimuli have a similar degree of perceptual invariance. Critically, robust stimuli are slightly more distorted via MSE than texform stimuli suggesting that the adversarially trained model has learned to represent peripheral computation better than the texform model by maximizing the perceptual null space and throwing away more useless low-level image features (hence achieving greater Mean Square Error).

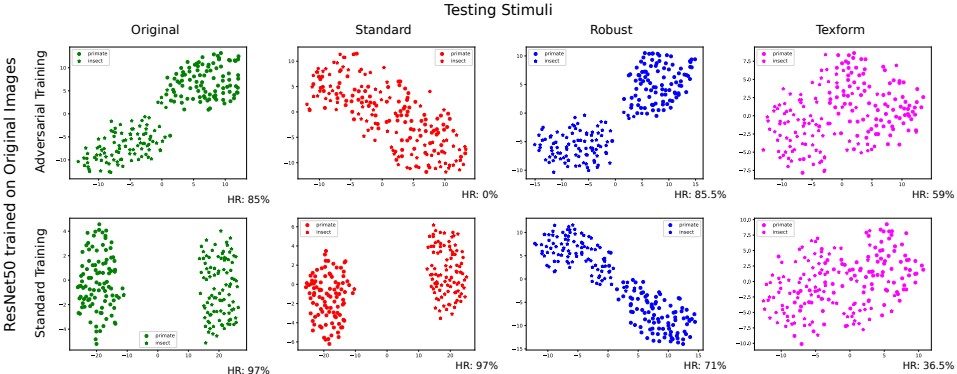

Figure 9: Here we show a 2D projection using t-SNE (Van der Maaten & Hinton, 2008) to visualize the outputs of the last layer of the Adversarially trained network (that was used to synthesize the Robust Stimuli), and the Standard trained network (that was used to synthesize the Standard stimuli), both on a family of different stimuli: Original, Standard, Robust and Texform. The Adversarially trained network – similar to the human – can not distinguish between 2-class Standard Stimuli (unlike the Standard Network that has a near perfect 2-class hit rate). Most importantly, the Adversarially trained network yields a near double hit rate on Texform classification wrt the Standard trained network. This suggests that the Adversarially trained network has a representation that is more perceptually aligned to models of Peripheral Computation than the Standard trained model.

