# OpenReview forum: "Finding Biological Plausibility for Adversarially Robust Features via Metameric Tasks"
_NeurIPS.cc/2021/Workshop/SVRHM — SVRHM 2021 Poster_

### Official Review · Reviewer_B5zo · 2021-10-19
**Missing methodological details and flawed logic**

**Rating:** 3
**Confidence:** 4

**Review:**

The authors report an experiment that compares the discriminability of between standard, robust, textform and original images in peripheral vision using two different tasks.  They find that participants find it easy to discriminate original from standard images at all peripheral locations, but find similar pattern of results for the Robust and textform stimuli.  The authors take these findings to support the conclusion that adversarially robust networks capture something important about peripheral vision.

There are a number of problems with this work that lead me to suggest rejection.  Methodologically there is almost no detail about the experiment.  For example, how many trials did each participant see?  And why only 12 participants tested?  This seems a small number of participants given that the authors want to make claims about adversarially robust networks on the basis that the results do not differ from the textform stimuli.  Unless there were many trials per participant, the study is likely to be underpowered to test for differences, and indeed, there are some reasonably large differences in perform between the robust vs. texform results in the 2AFC if I’m reading Figure 4 correctly.  The authors do not provide any information about how they insured that participants maintained fixation, was eye-tracking used or some behavioural task used to assess where participants eyes were looking?

But I think there may be a more fundamental conceptual flaw in the experiment that undermines their conclusion. The Standard images are clearly very easy to distinguish between from the original objects, so not surprising that performance was at ceiling at all peripheral conditions.  By contrast, the Robust and textform stimuli share obvious similarities to the original stimuli, and the general decline in performance with greater eccentricities is not surprising and would be expected on any stimulus that is somewhat similar to the original.  For example, if you simply blurred the original stimuli I expect you might get the same results, but would you want to say that a network that simply blurs stimuli is capturing something important about peripheral vision?  It is far too weak a test to make the conclusions that the authors are making.

Minor points:

Why rotate the adversarial images?
I find the paper unnecessarily had to follow in places.

---

### Official Review · Reviewer_uLiW · 2021-10-29
**Review of "Finding Biological Plausibility for Adversarially Robust Features via Metameric Tasks"**

**Rating:** 7
**Confidence:** 4

**Review:**

This paper investigates whether the visual periphery plays a role in robust processing of visual stimuli in humans vs machines. The authors run experiments in which human subjects are shown synthesized images which are generated by neural network models that are either 'standard' or 'adversarially robust', but the human subjects only see these images in their visual periphery, and perform a discrimination task. The authors conclude that there is a significant difference in standard images vs adversarially robust images, and therefore the models that generate the latter set of images perform peripheral computation analogous to the human visual processing system.

PROS

-The problem statement is interesting, as not much attention is given to foveal vs peripheral vision in humans vs machines. The authors are subtly hinting to move SOTA in computer vision towards incorporating peripheral vision by first studying the behavioral similarities/differences in humans vs machines.

-The authors assess image discriminability between the different models of interest, and base their results largely on the behavior of their human subjects. This makes their results relevant to the theme of the workshop.

CONS

-The authors show that perceptual discriminability doesn't decrease when participants are discriminating between 'original vs standard synthesized' images as retinal eccentricity increases, but it seems that the standard synthesized images look nothing like the original in Figure 2. This could be an experimental flaw, and possibly why discriminability stays fixed as retinal eccentricity increases.

-The authors make an implicit assumption that if percetual discriminability in their experiments decreases as a function of retinal eccentricity, then the corresponding model that synthesized those images is akin to human visual system. They don't explain exactly why this is, although they cite previous work on 'crowding' (Balas et al. (2009)). I would like to see more details about their logic in the main text.

(Details: The author state "If the decay rates of perceptual discriminability are similar across stimuli, then it suggests that the transformations learned in an adversarially trained network are isomorphic to the transformations done by models of peripheral computation - and thus, to the human visual system.")

-The authors make the claim "we found that adversarially trained networks encode a similar class of transformations that occur in the visual periphery", but this is quite a strong claim to make since the exact neuronal mechanisms of peripheral computation in the human visual system are not well-understood (AFAIK). The relationship between perceptual discriminability in peripheral vision in humans vs synthesized images may be analogous (which is exactly what the authors showed), but that doesn't imply the transformations are necessarily similar.

---

### Official Review · Reviewer_a74s · 2021-10-29
**A mature project with potentially important implications. Should be considered for highlighting at SVRHM.**

**Rating:** 8
**Confidence:** 4

**Review:**

_I was happened to be assigned with this work also in ICLR. I believe the back and forth will take place there but I include my review here again to support the SVRHM decision process._

This study draws a surprising connection between adversarially trained CNNs and human peripheral perception. Using the synthesized metamers, the authors show that the human ability to discriminate between natural images and their synthetic metamers drops in a similar fashion for metamers generated by inverting adversarially trained CNNs and for metamers generated by an inversion of a well-studied model of peripheral vision (i.e., "Texforms") as display eccentricity increases. This may indicate that adversarially trained models bear some resemblance to human visual processing at the retinal periphery.

I enjoyed reading the paper and found the connections it makes inspiring.

Thoughts:
* Overall, the metameric manipulation seems to make a simple prediction: if a model correctly captures human vision for a given eccentricity, then the metamers should not be discriminable by humans when presented at that eccentricity. The authors interpret the decreasing discriminability as indicative that robust CNNs enjoy a similar relation to peripheral vision as the Freeman & Simoncelli model does. I agree that this is a reasonable interpretation. However, the data also suggests a discrepancy between robust CNNs and peripheral vision, as discriminability remains above chance even for 30 visual degrees. The authors might consider addressing this discrepancy in the text and in subsequent analyses, analyzing what humans see that the model can't.
* Doesn't fitting the two free parameters of the Freeman & Simoncelli model to the robust CNN weaken the conclusion that the two models are equivalent in their predictive power? What if the Texforms are formed independently of the CNN?
* It is unclear to me whether it's fair to generate metamers to be tested in the periphery without simulating visual acuity limitations during stimulus synthesis. The standard CNN might be driven by high-frequency information that isn't even there when the stimulus is presented at 20 or 30 degrees. At least, the authors might want to evaluate posthoc whether the metamers remain metamers under high-pass filtering emulating retinal constraints at the different eccentricities.
* The synthetic vs. synthetic condition is included without sufficient motivation or discussion. How should this condition be interpreted differently than the natural vs. synthetic condition? This point is particularly relevant for the standard CNN where these two conditions diverge.
* Do subject demographics, ethical approval, and subject compensation appear in the text?
* Open-sourcing the code, stimuli, and behavioral results would increase the impact of this work.
* Learning invariances shared between foveal and peripheral processing is an interesting idea! Perhaps this approach can replace adversarial training. This direction should be pursued further in another study.

Minor points:

* The discussion treats standard CNNs as spatially uniform. However, in practice, this is not the case due to effects related to pooling (Azulay & Weiss, 2018) and padding (Alsallakh et al., 2020). The latter study even indicates implicit foveation.

* Some of the cited ArXiv citations are outdated and should be replaced with citations of the corresponding conference proceedings.

References: Alsallakh, B., Kokhlikyan, N., Miglani, V., Yuan, J., & Reblitz-Richardson, O. (2020). Mind the Pad--CNNs can Develop Blind Spots. arXiv preprint arXiv:2010.02178.

Azulay, Aharon, and Yair Weiss. "Why do deep convolutional networks generalize so poorly to small image transformations?." arXiv preprint arXiv:1805.12177 (2018).

---

### Decision · Program_Chairs · 2021-11-02

Accept (Poster)